# ODE-based Recurrent Model-free Reinforcement Learning for POMDPs

**Xuanle Zhao[1, 2], Duzhen Zhang[1, 2], Liyuan Han[1, 2], Tielin Zhang[1, 2],\* Bo Xu[1, 2, 3]**

[1]Institute of Automation, Chinese Academy of Sciences, Beijing, China
[2]School of Artificial Intelligence, University of Chinese Academy of Sciences, Beijing, China
[3]Center for Excellence in Brain Science and Intelligence Technology,
Chinese Academy of Sciences, Shanghai, China

{zhaoxuanle2022, zhangduzhen2019, hanliyuan2019, tielin.zhang, xubo}@ia.ac.cn

## Abstract

Neural ordinary differential equations (ODEs) are widely recognized as the standard for modeling physical mechanisms, which help to perform approximate inference in unknown physical or biological environments. In partially observable (PO) environments, how to infer unseen information from raw observations puzzled the agents. By using a recurrent policy with a compact context, context-based reinforcement learning provides a flexible way to extract unobservable information from historical transitions. To help the agent extract more dynamics-related information, we present a novel ODE-based recurrent model combined with model-free reinforcement learning (RL) framework to solve partially observable Markov decision processes (POMDPs). We experimentally demonstrate the efficacy of our methods across various PO continuous control and meta-RL tasks. Furthermore, our experiments illustrate that our method is robust against irregular observations, owing to the ability of ODEs to model irregularly-sampled time series.

## 1 Introduction

Conventional reinforcement learning (RL) is typically cast as a problem of solving fully observable Markov decision process (MDP) tasks which are trained and tested on the same task. However, most practical applications require the agents to handle some degree of partially observable and irregular observations. Humans are always good at solving these kinds of tasks by extracting crucial information from past observations and actions, while conventional RL agents do not have the ability to extract information relevant to the tasks.

In recent works, several categories of methods have been proposed to solve PO problems. The most straightforward one is to include all historical observations as input [Lee et al., 2020]. However, this kind of method is impractical in application owing to the dimension increment of the observation inputs. Another category is based on model-free methods which generally use recurrent neural networks (RNNs) as memory units and function approximators to extract historical task-related information to solve partial observable problems [Igl et al., 2018, Kapturowski et al., 2019, Han et al., 2019, Jaderberg et al., 2019]. Recurrent neural networks are commonly employed to encode historical state transitions into contextual representations, leveraging their capacity to process sequential information. These representations are then used in conjunction with the current state to find the optimal policy. This baseline is simple and efficient, as it obtains near-optimal results and only requires changing the policy and value network into a recurrent version. The third category considers using a model to estimate a belief state from historical transitions [Lee et al., 2020]. Agent receives

---

*Corresponding author

37th Conference on Neural Information Processing Systems (NeurIPS 2023).

belief states as observations that contain useful information to solve the tasks, this category is much similar to the second one. However, almost all notable breakthroughs have been showcased in discrete-time problems such as games and discrete control problems [Mnih et al., 2015, Silver et al., 2016], whereas most physical and biological systems in the real world are inherently continuous in time and follow differential equation dynamics.

Differential equations are commonly considered the benchmark for modeling physical mechanisms [Schölkopf et al., 2021]. They permit people to forecast the future behaviors of physical systems and the statistical interdependencies among variables. Moreover, they provide physical insights, explain the system's functioning, and justify reasons for their causal structure [Peters et al., 2022, Zhang et al., 2023b]. A discretized control system will converge to an ODE as the time increment approaches zero. However, there is no consensus on how to choose the right discretization stepsizes and account for irregular observation times. In current paradigms, stepsizes are fixed in advance and vary in environments. For example, in classic control tasks (such as CartPole and Pendulum), observations are sampled by regularly discretized timesteps [Brockman et al., 2016].

In this study, similar to context-based RL, we developed an ODE-based recurrent model (GRU-ODE) that encodes historical observations, actions, and rewards into a latent variable. Previous works have shown that ODE-based recurrent models could simulate the physical dynamic system in an auto-regressive fashion, such as Latent-ODE [Rubanova et al., 2019] and ODE-LSTM [Lechner and Hasani, 2020]. The GRU-ODE is based on recurrent networks, which modify the GRU topology by separating its memory cell into time-continuous states. At each time step, the model first iterates using the conventional GRU equations and then computing the differentiation based on the ODEs [Chen et al., 2018]. We then propose a method to solve PO tasks by training the GRU-ODE and the actor-critic algorithm. We show owing to the differential process, our model could extract more accurate unobservable information from partially observable environments. Our approach outperforms baselines in several regular PO and Meta-RL control tasks and irregular PO tasks.

## 2    Preliminary and Related work

Typical POMDPs are used to describe decision or control problems in which the underlying states of the environments cannot be directly observed. A POMDP is a tuple $(\mathcal{S}, \mathcal{A}, \mathcal{O}, T, O, R, \gamma)$. $\mathcal{S}$ is a set of states, $\mathcal{A}$ is a set of actions, and $T : \mathcal{S} \times \mathcal{A}$ is the state-transition probability function. Let $\mathcal{O}$ be the observation set and let $O : \mathcal{S} \times \mathcal{A} \times \mathcal{O} \to [0, 1]$ be the observation probability. The reward function $R : \mathcal{S} \times \mathcal{A}$ decides the reward during state transition and $\gamma$ is the discount factor. The objective is to learn a policy function to maximize expected discounted rewards $\mathbb{E}\left[\sum_{t=0}^{T-1} \gamma^t r_{t+1}\right]$ with a finite horizon $T$.

Common PO tasks include scenarios of partially occluded states [Heess et al., 2015], randomly dropped frames [Hausknecht and Stone, 2015], egocentric images [Zhu et al., 2017], and randomly noised observations [Meng et al., 2021]. These kinds of problems are hard to solve because the dimension of historical observations grows linearly with timesteps. Prior works consider learning a belief state to encode underlying environments by extracting information from historical state transitions [Zintgraf et al., 2019]. Some other methods use memory-based policy, which takes the entire history states as inputs [Lee et al., 2020]. Recent works have widely used recurrent models to equip relevant algorithms [Ni et al., 2022, Han et al., 2019]. These recurrent methods could be further divided into model-free and model-based methods according to whether their training objectives contain transition functions.

In the meta-RL setting, some indicator of the task is unobserved and methods are quite similar to that of PO tasks. Prominent approaches utilize recurrent networks for fast adaption [Duan et al., 2016]. Recent studies demonstrate that the recurrent model-free algorithm serves as a robust baseline for meta-RL [Ni et al., 2022]. This is accomplished by equipping the policy network with a recurrent model, which aggregates past state and action transitions into an auxiliary context input [Zintgraf et al., 2019]. Recent works also consider training non-recurrent models to encode the context [Mu et al., 2022, Guo et al., 2022]. These approaches can be categorized into context-based meta-RL methods. Another category uses policy gradients [Finn et al., 2017, Xu et al., 2018a] or hyperparameters [Xu et al., 2018b] to learn from aggregated experience are defined as gradient-based methods.

Neural ODEs have been widely used to tackle irregular time series [Chen et al., 2018]. Standard RNN treats observations as a sequence of tokens and does not account for variable timestep between observations. When facing irregular-sized data, another interpolation or generative adversarial network is used to perform interpolation and imputation. However, these methods are not robust in unseen environments. Chen et al. [2018] use residual neural networks with time-invariant dynamics to solve the ODE initial-value problem (IVP). Recent works build continuous time models by adding ODEs to the recurrent cells update process [De Brouwer et al., 2019, Kidger et al., 2020]. CT-LSTM [Mei and Eisner, 2017] combines the LSTM architecture with the continuous-time neural Hawkes process. ODE-LSTM [Lechner and Hasani, 2020] transforms the internal dynamical flow of LSTM to a continuous-time model and demonstrates its efficacy in learning kinematics simulation. Liquid time-constant networks (LTC) [Hasani et al., 2021, Lechner et al., 2020] represent dynamic systems with liquid time constants coupled to their hidden states and compute outputs by numerical differential equation solvers.

Recent studies in scientific machine learning have attempted to utilize dynamic encoding or physical mechanisms within their respective fields, such as DeepONet [Lu et al., 2019] and PINNs [Cuomo et al., 2022, Krishnapriyan et al., 2021]. Incorporating physical insights has been shown to enhance the efficiency of neural networks and the ability of representation learning [Zhang et al., 2022, Yang et al., 2021, Han et al., 2023, Zhang et al., 2023a]. Some previous works combine ODEs with model-based RL. Du et al. [2020] use Latent-ODE to learn the transition model to solve semi-Markov decision processes (SMDPs), which also shows variables encoded by Latent-ODE capture unobservable state representations in PO environment. Yildiz et al. [2021] infer the unknown state evolution differentials with Bayesian neural ODEs. Some other works use ODEs to learn the dynamic or transition functions and apply them to continuous and irregular time interval tasks [Salehi et al., 2022, Ainsworth et al., 2021].

## 3 Approach

This section first describes constructing ODE-based recurrent models to encode historical information into context variables for model-free RL. Then, we describe the overall training objective and how to use our model to optimize the policy network and value function.

In the typical POMDP setting, the reward and transition functions share some common structures across the MDPs. Some embeddings must represent the detail of the task information, which could not be accessed in advance. Context-based RL methods attempt to solve partial observable problems by allowing the policy network to receive an auxiliary context variable which includes historical information. The context variable at time step $t$ is inferred from the agents' experience up to the current states,

$$\tau_{:t} = (o_0, a_0, r_1, o_1, a_1, \cdots, o_{t-1}, a_{t-1}, r_t, o_t). \tag{1}$$

There should be a superscript $i$ to denote various episodes, we drop it for ease of notation. In POMDP environments, observations are denoted as $o_t$ instead of $s_t$, as the underlying states are only partially observable and must be inferred based on the available observations. Based on the above $\tau_{:t}$, it is sufficient to encode historical information into a context variable $z_t$, rather than modeling transition and reward dynamics which consist of millions of parameters. Generally, the memory-based policy is defined as $\pi(a_t|\tau_{:t})$, conditioning on the whole history.

### 3.1 Model definition

Following ODE-RNN [Rubanova et al., 2019], we extend hidden state transitions in RNNs to continuous-time dynamics defined by Neural ODEs. As RNNs with exponentially-decayed hidden state follow an implicit ODE of the form $\frac{dh(t)}{dt} = -\tau h$ with $h(t_0) = h_0$, where $\tau$ is a parameter, the hidden states could be modeled by Neural ODEs. Therefore, we propose our ODE-based recurrent model.

**GRU-ODE.** In our models, transformations of hidden states and transitions between latent states are computed using recurrent cells and neural ODEs, respectively. The resulting model is

$$\tilde{h}_t = \text{GRUCell}\left(h_{t-1}, x_t\right),$$
$$h_t = \text{ODESolve}\left(f_\theta, \tilde{h}_t, dt\right), \qquad (2)$$
$$z_t = \mathcal{N}(\mu_{h_t}, \sigma_{h_t}) = o(h_t),$$

where the context variable $z_t$ is defined as a Gaussian distribution parameterized by mean $\mu_{h_t}$ and standard deviation $\sigma_{h_t}$. $x_t$ is the concatenation of $o_t$, $a_{t-1}$ and $r_t$. The gate of the *GRUCell* is controlled by the output of *ODESolve* $h_t$, instead of its own output $\tilde{h}_t$. The *ODESolve* are numerical differential solvers implemented by neural networks

$$\text{ODESolve}\left(f_\theta, h(t), dt\right) = h(t) + \int_t^{t+dt} f_\theta(h(t))dt, \qquad (3)$$

where $f_\theta$ computes the differential of $h(t)$ which are implemented by neural networks. The time increment $dt$ in the numerical ODE solver is fixed for conventional control tasks and varied in irregular observation tasks. The input of our model $x_t$ is the concatenation of observations $o_t$, actions $a_{t-1}$ and rewards $r_t$. We found that the reward signals could help the agent learn unobservable information, leading to improved agent performance in PO environments. As the reward function typically consists of $r(s,a) = d(s, s^*) + c||a||_2$, where the first term measures the distance between the current state and the objective state and the latter term contains the penalty of the action magnitude. Therefore, the unobservable state information could be inferred based on tuples of observations, actions, and rewards. To verify the rationality of our proposed model, we simplify the classic control task. In PO dynamic systems, the underlying physics of the task is always supposed to be second-order, the observable states are expressed in terms of either position $\theta$ or angular velocity $\omega$. As the state and action space could be denoted as $s(t) = [\theta(t), \omega(t)]^T$ and $a(t)$ respectively, the dynamic system with explicit control is defined as follows:

$$\frac{d\theta(t)}{dt} = \omega(t), \quad \frac{d\omega(t)}{dt} = \mathcal{A}(\theta(t), \omega(t), a(t)), \qquad (4)$$

where $\mathcal{A}(\cdot)$ is referred to as acceleration field. Simplifying the update equation, this procedure could be modeled as differential equations with explicit actions. The current state could be modeled using *ODESolve* with initial state as $s(t) = s(0) + \int_0^t f_\theta(s(\tau), a(\tau))d\tau$, which could be solved by neural ODEs [Chen et al., 2018]. Previous neural ODE models were always used for time-series predictions, which do not contain explicit actions. Chiappa et al. [2017] incorporate actions as auxiliary inputs of recurrent cells to build recurrent simulators. We combine them to build our recurrent models. In addition to states, actions and rewards can also be solved as initial value problems.

**Proposition 1.** *The action $a(t)$ is differential with time like the state $s(t)$, as the action is controlled by a deterministic policy function $\pi(\cdot)$. The derivative of $a(t)$ with respect to time $t$ could be expressed as*

$$\frac{da(t)}{dt} = \frac{d\pi(s(t))}{dt} = \frac{d\pi(s(t))}{ds(t)} \frac{ds(t)}{dt}. \qquad (5)$$

The policy function $\pi(\cdot)$ is implemented with neural networks, so $d\pi(s(t))/ds(t)$ is the differential between the output and input of the policy network and could be computed by the chain rule. Following the above formulation, the reward function is also differentiable as it is always the linear combination of state $s(t)$ and action $a(t)$. Therefore, the update process of GRU-ODE input $x(t)$, the concatenation of state, action and reward, could be modeled as ODEs and solved by *ODESolve*. As the latent context variable $z_t$ reflects the general dynamic across different episodes, we reckon the update process of $z_t$ could be modeled in the same way with explicit input $x(t)$. Our model explicitly changes the discrete update into a continuous case, which represents latent variables more precisely. Existing implementations of state transitions perform inaccurate numerical integration routines, which results in rough estimation. For example, in the CartPole task crude numerical solver lead approximations bifurcate near the pole hangs up positions [Yildiz et al., 2021]. The accuracy of the approximation is dependent on the stepsize of the solver. Consequently, continuous methods tend to provide a closer approximation to the true value.

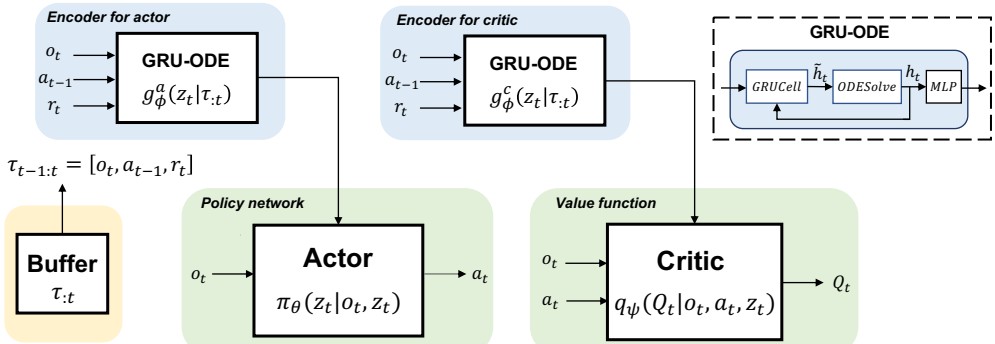

Figure 1: Network architecture. Trajectories of observations, actions, and rewards are sampled and stored in the buffer. During the training procedure, $\tau_{t-1:t}$ is processed online using a GRUCell to produce the embedding $\tilde{h}_t$. Subsequently, $\tilde{h}_t$ is passed through an ODE solver and MLP to encode the context variable $z_t$. We use separate encoders for the actor and critic networks to improve the performance.

**Proposition 2.** *The global and local truncation errors of the Euler numerical solver are $O(h)$ and $O(h^2)$ respectively. As the stepsize $h$ approaches zero, the truncation error tends to $\lim_{h \to 0} |O(h)| = 0$.*
Assuming the increment $dt$ in the ODEs equals the observation interval, the discretized system will converge to the actual ODE as the time increment approaches zero.

Similar to relevant context-based methods, we use a recurrent ODE-based model to encode past trajectories into a variable. Compared with ODE-based models with encoder-decoder structures, models transformed from typical recurrent models could predict online at each time step. As opposed to standard RNNs, ODE-based RNNs learn the dynamics between observations, rather than having them pre-defined, which enables handling irregular data and modeling continuous sequences without relying on pre-assumptions about the dynamics of the observation sequences [Rubanova et al., 2019]. Moreover, we found that owing to their underlying physical principles, ODEs are suitable for inferring unknown physical information, which is significant in partially observable environments.

### 3.2 Training procedure

Now we have the GRU-ODE model to represent the context variable, we move on to the question of identifying learning procedure and optimal policy. As mentioned above, with partial observations, latent context $z_t$ is used to provide historically unknown information. Prior works have shown using separate recurrent encoders for the actor and critic could achieve high rewards [Meng et al., 2021]. Ni et al. [2022] show the gradient norms of recurrent context encoders vary in the actor and critic resulting in the gradient of critic loss dominating the actors', which will impact the training process of the actor's context encoder. Therefore, we use separate GRU-ODE models for context representation and implement them into Actor-Critic algorithms to improve the final performance.

**Model implementation.** The general training objective maximizes the discounted reward in an episode. We implement our method based on TD3 Fujimoto et al. [2018] and SAC Haarnoja et al. [2018], so the policy network and value function should be parameterized differently. Deep neural networks are used to represent individual components. There are :

1. The recurrent context encoder GRU-ODE model $g_\phi(z_t|\tau_{:t})$, parameterised by $\phi$. We use superscripts $a$ or $c$ to denote the encoder and context variables used for the policy network or the value function.

2. The conditioned policy network (actor) $\pi_\theta(a_t|o_t, g_\phi^a(z_t^a|\tau_{:t}))$ parameterized by $\theta$ with context variable $z_t^a$.

3. The conditioned value function (critic) $q_\psi(Q_t|o_t, a_t, g_\phi^c(z_t^c|\tau_{:t}))$ parameterized by $\phi$ with context variable $z_t^c$.

The network architecture is shown in Fig. 1. The context variable $z_t$ is regarded as a posterior and represented by the distribution's parameters, the details are discussed later. Compared to modeling the transition or reward function directly using the network, learning distribution embeddings need fewer parameters which makes inference easier.

**Training objectives.** The conventional training objective of the policy network $\pi_\theta(\cdot)$ is to maximize the expected return

$$\mathcal{J}(\theta) = \mathbb{E}_{\pi_\theta} \left[ \sum_{t=0}^{T} \gamma^t R(r_{t+1}|s_t, a_t) \right]. \tag{6}$$

After experimental analysis, it was discovered that utilizing $\mathcal{J}(\theta)$ directly in conditioned policy network and value function with recurrent encoders results in significant variation of the embedding variable $z_t$ within an episode. This variation can lead to a degradation of the context encoder's representation ability in PO environments. Therefore, we adjust the original training objective with an embedding constraint term to regularize $z_t$. Now, the overall training objective is to maximize

$$\mathcal{L}(\phi, \theta, \psi) = \mathbb{E}_\rho \left[ \mathcal{J}(\psi, \phi, \theta) - \lambda \mathcal{K}(\phi) \right], \tag{7}$$

where $\rho$ denotes the trajectory distribution induced by the sample policy. Expectations are approximated by Monte Carlo samples. Conditioned delayed policy update algorithms are implemented in our method, thus $\mathcal{J}(\psi, \phi, \theta)$ are used to denote actor and critic networks with different parameters. To regularize the context variable $z_t$, we add the Kullback–Leibler (KL) divergence $\mathcal{K}(\phi)$ between posterior and prior distributions

$$\mathcal{K}(\phi) = \sum_{t=0}^{T} D_{KL} \left( g_\phi(z_t|\tau_{:t}) || p(z_t) \right). \tag{8}$$

We set the prior to our previous posterior, $g_\phi(z_{t-1}|\tau_{:t-1})$, with initial prior $g_\phi(z_0) = \mathcal{N}(0, I)$. $D_{KL}$ denotes the KL divergence, which could be optimized by the reparameterization trick. Since both terms in $\mathcal{K}(\phi)$ are parameterized by Gaussian distributions, the KL-divergence can be analytically calculated as

$$D_{KL} \left( g_\phi(z_t|\tau_{:t}) || p(z_t) \right) = \log \frac{\sigma_{\phi,t}}{\sigma_{\phi,t-1}} + \frac{(\mu_{\phi,t} - \mu_{\phi,t-1})^2 + \sigma_{\phi,t}^2}{2\sigma_{\phi,t-1}^2} - \frac{1}{2} \tag{9}$$

In this work, past trajectories are encoded by GRU-ODE. Eq. 7 is optimized end-to-end, and $\lambda$ is the balance weight that supervises the encoder learning objective against the RL objective $\mathcal{J}(\psi, \phi, \theta)$, which is objective to find an optimal policy and accurate value function. To improve the efficiency of the training process, we sample fixed-length truncated sequences as mini-batches for training our model, rather than using the whole episodes.

Previous work mentioned that training the encoder and the policy network separately using data from different buffers can prevent the gradients of opposing losses from interfering with each other. Zintgraf et al. [2019] follow this setting to pre-train the encoder in advance before the policy network. We found this setting is unnecessary in practice, the data buffer is shared in our method.

## 4 Experiments

We experimentally evaluate our models across several PO regular/irregular observation domains. In regular observation domains, we consider conventional partially observable control and meta-RL tasks by employing MuJoCo [Todorov et al., 2012] and PyBullet [Greff et al., 2022] environments. In irregular observation domains, we consider modifying classical environments: Pendulum and CartPole tasks[Brockman et al., 2016] into irregular time intervals with surrogate rewards.

### 4.1 Partially Observable Classic Control Tasks

The Pendulum and CartPole tasks are classic control tasks for evaluating RL algorithms. For the Pendulum task, the goal is to learn a policy to swing the pendulum up and maintain it at the highest position to obtain more rewards. For the CartPole task, the goal is to learn a policy to preclude the pole from falling down and forestall the cart from running away by exerting force on the cart.

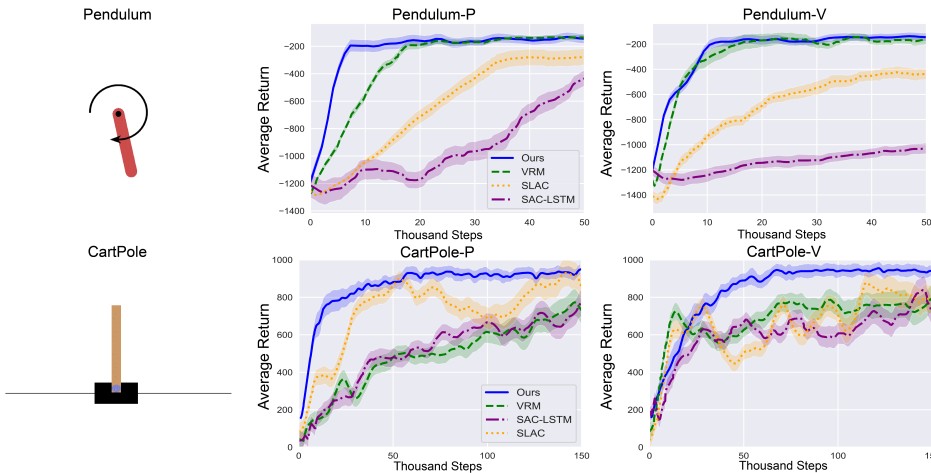

Figure 2: Classic partial observable control tasks. The shaded region represents a standard deviation of average evaluation over five runs and Curves are smoothed for visual clarity.

The observable information consists of the coordinates of the cart, the angle of the pole, and their velocities.

These classic control tasks are relatively easy to solve in fully observable domains. Therefore, these PO tasks can effectively underscore the problem of representation learning. Experiments are performed in PO cases of these two control tasks, in which only velocities or positions could be observed (we use the suffix -V/-P to name). The PO setting is more practically significant, as in many real-world applications, agents may only be able to estimate partial state information.

We compare our method with SLAC, VRM, and SAC-LSTM. SLAC [Lee et al., 2020] combines off-policy model-free RL with representation learning via a sequential stochastic state space model. SLAC was developed for pixel observations, we follow the modifications in Han et al. [2019] to compare it with our method. VRM [Han et al., 2019] is a recent, model-based POMDP algorithm, which uses variational recurrent models to encode context variables and separate representation learning from dynamic programming. In the SAC-LSTM, soft actor-critic algorithms with recurrent networks are used as function approximators. We follow the settings in [Han et al., 2019] to construct the above methods.

As expected, our method succeeds in learning to solve all these tasks and performs better than VRM which is the state-of-the-art method for solving POMDP tasks. While SLAC performs well in the CartPole tasks and less sample efficiency in Pendulum tasks. SAC-LSTM needs more epochs for training and fails to solve the Pendulum-V task. We reckon these results are attributed to the ability of ODEs to model physical systems and infer unknown information.

To assess the capability of our model to deduce unobservable environmental information, we project the latent variable $z_t$ in an episode to 2D using UMAP [McInnes et al., 2018]. We estimate the unobservable state information such as the angular velocities in the Pendulum-P task through observations, actions, and rewards (as the rewards are calculated based on state and actions, the unobservable state could be computed in reverse), and scatter plot color based on the logarithmic scaling

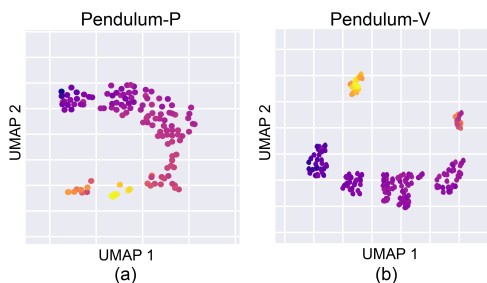

Figure 3: The visualization of the latent variables $z_t$ in one episode after dimension reduction. Points are colored according to (a) the logarithm of the calculated angular velocity and (b) the logarithm of the calculated angular position (angle difference from the initial position). Each figure consists of 201 points including the initialized Gaussian $z_0$, as the maximum step in the Pendulum task is 200.

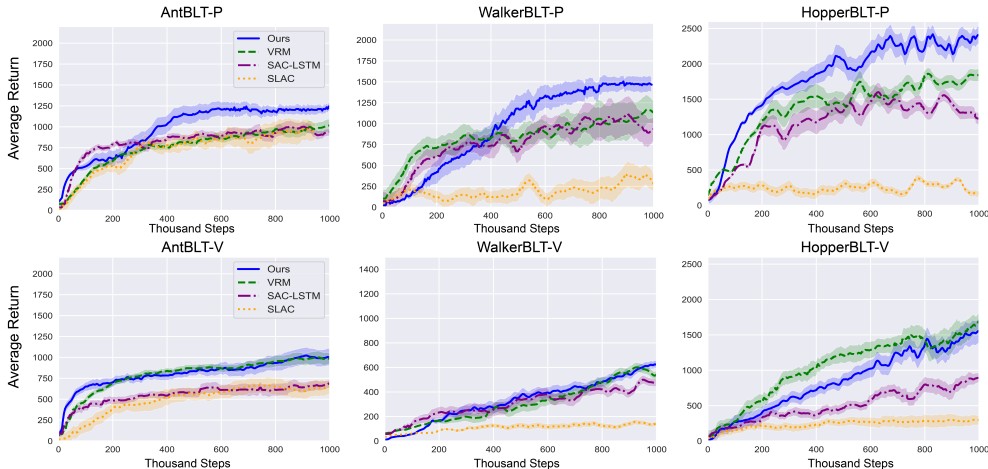

Figure 4: Learning curve of the partially observable PyBullet robotic control tasks, plotted in the same way as in Fig. 2. The PyBullet environments are ported from deprecated Roboschool environments which are harder than the MuJoCo Gym environments[1].

of their values. Fig. 3 shows the clustering of latent variables corresponds closely to the unobservable physical parameters, which demonstrates the efficacy of our GRU-ODE context encoder.

## 4.2 Partially Observable Robotic Control Tasks

We evaluate the performance of our proposed method in the PyBullet control environments [Greff et al., 2022] which are more challenging than MuJoCo. The PyBullet environments contain several continuous robotic control tasks and the state of fully observable environments includes the robot's co-ordinates, trigonometric functions of joint angles, and angular velocities. We modify the observations into PO versions with the same criteria and compare them with the same methods as the PO classic control tasks. As experimental results demonstrated our method obtains policy improvement in the majority of PO robotic control tasks, especially in environments where only the position information is observable. For tasks with only positions observable, our method outperforms the other three methods and presumably shows efficiency in extracting useful information from historical position observations. We reckon that velocity could be simply estimated by one-step differentiation in coordinates and joint angles of the robot, which eases representation and policy learning. Environments with only velocities that could be observed are much harder to solve. However, deducing the position information solely from the given velocity is not practical. In environments where only velocity can be observed, our method performs similarly to VRM. Additionally, our results indicate that the learning processes of the SLAC method exhibited instability, where it intermittently achieved a nearly optimal policy, but frequently converged to a suboptimal one. As a result, the average performance of SLAC was generally less favorable than that of our approach across the majority of PO robotic control tasks. SAC-LSTM fails to solve almost all these tasks.

We further compare our methods with two more baselines RMF (Recurrent Model-Free)[Ni et al., 2022] and TD3-FPOW (TD3 with fixed previous observation window). The complete results are shown in Table 6.

## 4.3 Meta-learning MuJoCo Control Tasks

We further show our method can also scale to other types of PO tasks, such as Meta-RL, by applying our method to a diverse set of domains, such as point robot and MuJuCo locomotion tasks, which are commonly used in meta-RL literature. Our tasks involve utilizing the semi-circle, which refers to a point robot that navigates along a semi-circular path in search of a sparsely located reward, as well as wind tasks, where a point robot navigates to a fixed goal amidst fluctuating wind conditions. For MuJuCo locomotion tasks, we consider using the HalfCheetah-Dir environment where the agent has

---

[1]https://github.com/openai/roboschool#deprecated-please-use-pybullet-instead

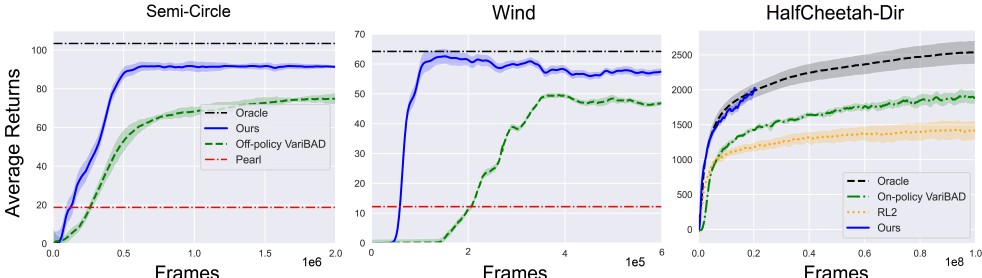

Figure 5: Learning curves with compared methods on meta-RL tasks. Unlike other methods, we only plot the final return of Oracle and Pearl methods in the semi-circle and wind tasks. The learning curve data of on-policy VariBAD, Oracle, and RL2 in the HalfCheetah-Dir task are copied from the public GitHub repository of variBAD[2].

to control two legs run either forward or backward. The MuJuCo-relevant tasks are much harder than the prior point robot tasks.

We compare our method with various methods, such as VariBAD, RL2 [Duan et al., 2016], Pearl [Rakelly et al., 2019], and Oracle policy. VariBAD utilizes a variational, model-based objective to learn task embeddings explicitly. The originally proposed variBAD [Zintgraf et al., 2019] uses PPO [Schulman et al., 2017], and the same method was implemented with SAC [Dorfman et al., 2020]. These two methods could be named on-policy variBAD and off-policy variBAD. An oracle policy can access the POMDP hidden state to transform the POMDP into an MDP, thus this policy could be treated as an upper bound on the performance of related methods. We follow the settings in Ni et al. [2022] to construct the above methods.

Fig. 5 shows that our method outperforms off-policy variBAD and Pearl in the semi-circle and wind environments, and on-policy variBAD and RL2 in the HalfCheetah-Dir environment, both in terms of sample efficiency and asymptotic return. As our model is trained end-to-end, without using pre-trained encoders to represent task contexts like off-policy variBAD, which do not suffer out-of-date representation problems. On-policy variBAD uses autoencoder architecture to reconstruct the state and reward, which are more advantageous to context representation. Our method shows simple recurrent encoder could also stabilize representation learning and improve policy performance.

## 4.4 Temporally Irregular Observable Control Tasks

In standard RL frameworks such as OpenAI Gym, the states/observations arrive at constant intervals as $\Delta t = \delta$. To contrast the robustness of our continuous-time method with other discrete POMDP methods, we evaluate related methods with irregularly sampled observations. We consider using an irregular sampling scenario in which observations arrive uniformly within a range $\Delta t \sim \mathcal{U}(0, 2\delta)$. We choose the mean time difference $\delta = 0.05$ for observation spacings, which is the standard interval setting of the Gym Pendulum task.

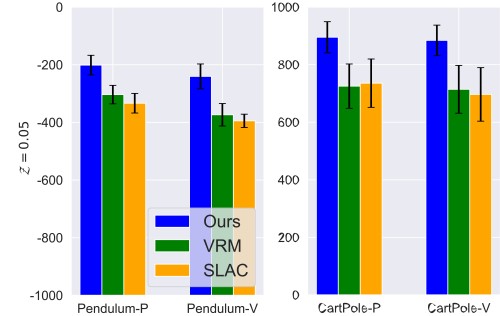

Figure 6: A comparison of PODMP methods with uniformly irregular observations.

Fig. 6 exhibits the final average returns with a standard deviation of VRM, SLAC, and our method in irregular Pendulum and CartPole tasks. The results show our ODE-based recurrent model is more robust with stochastic time increments, which is reasonable as ODEs build discrete-time sequences into continuous ones. This environment has more practical significance, as observation intervals are usually disturbed in practical application.

[2]https://github.com/lmzintgraf/varibad

## 5 Limitation

Although we have demonstrated that ODE-based encoders have advantages in robotic control tasks, their efficiency in discrete game problems, such as Atari, is not promising. In Atari, the states of the agents are not updated based on numerical differential equations, resulting in the inability to represent the state and action as integration processes. Also, the reward function is typically sparse and predefined. Implementing ODE-based encoders to model related processes does not make sense.

In addition, since the integration process is implemented by neural networks, the training procedure of ODE-based encoders requires more computation resources, which results in increased training time, especially for high-order numerical differential solvers. Although we find that using simple numerical solvers is enough in the majority of tasks (see Appendix Section 10), accelerating the integration process is still significant in solving differential equations and downstream tasks.

## 6 Conclusion and future works

We incorporate ODE-based recurrent neural networks with model-free reinforcement learning to solve POMDP control tasks. Through our empirical assessment across several partially observable domains, we showed that the policy network and value function conditioned on context variables encoded by GRU-ODE help the agents infer unknown observations and maximize expected returns. We experimentally demonstrate that our method is robust to environmental changes such as irregularity.

Reinforcement learning has been widely used in physical or biological systems to explain some of the mechanisms. However, some mechanisms in the brain are still unknown, such as how the brain arbitrates or allocates control over various sub-systems and the role of the prefrontal cortex and striatum in controlling. The classic leaky integrate and fire (LIF) neuron model is quite similar to ODEs if we ignore the firing process. We reckon combining ODEs with RL could help biologists explore mechanisms in the human brain.

## Acknowledgment

This work was supported by the Strategic Priority Research Program of Chinese Academy of Sciences (Grant No. XDA27010404), the Beijing Nova Program (Grant No. 20230484369), the Shanghai Municipal Science and Technology Major Project (Grant No. 2021SHZDZX), and the Youth Innovation Promotion Association of Chinese Academy of Sciences.

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

## Supplementary Material

In this supplementary material, we provide further information about our implementation details and ablation studies.

## 7 Truncation Error

The update procedure of the Euler Equation

$$y_{i+1} = y_i + hf(x_i, y_i), \tag{10}$$

and the accurate update equation is

$$y_{i+1} = y_i + hf(x_i, y(x_i)) + \frac{h^2}{2} f''(\xi_i), \quad \xi_i \in (x_i, x_{i+1}). \tag{11}$$

The lost item $\frac{h^2}{2} f''(\xi_i)$ is the local truncation error of the Euler method. The global truncation error is defined as

$$
\begin{aligned}
e_{i+1} &= y(x_{i+1}) - y(i+1) \\
&= (y_i + hf(x_i, y(x_i)) + \frac{h^2}{2} f''(\xi_i)) - (y_i + hf(x_i, y_i)) \\
&= e_i + h(f(x_i, y(x_i)) - f(x_i, y_i)) - \frac{h^2}{2} f''(\xi_i)
\end{aligned}
$$

In Euler method $f(x, y)$ satisfy the Lipschitz condition for $y$ and denote $T_{i+1} = -\frac{h^2}{2} f''(\xi_i)$ is the local truncation error and $T = \max\limits_{i} |T_i| = O(h^2)$

$$
\begin{aligned}
|e_{i+1}| &\le |e_i| + h|f(x_i, y(x_i)) - f(x_i, y_i)| + |T_{i+1}| \\
&\le |e_i| + h|y(x_i) - y_i| + |T_{i+1}| \\
&\le |e_i|(1 + hL) + T \\
&\le (1 + hL)^{i+1}(|e_0| + \frac{T}{hL}) \\
&\le e^{(i+1)hL}(|e_0| + \frac{T}{hL}) \\
&= O(h)
\end{aligned}
$$

where $e^0 = y(x - 0) - y_0 = 0$.

## 8 Enivronment Specification

We incorporate the occlusion benchmark proposed by VRM and replace the deprecated roboschool with PyBullet, as the official GitHub repository suggests. We transform the original MDP task into a POMDP version by removing all position/angle-related entries in the observation space for "-V" environments and velocity-related entries for "-P" environments.

**{Pendulum, CartPole, AntBLT, WalkerBLT, HopperBLT}-P.** The "-P" denotes environments that retain position-related observations by eliminating velocity-related observations.

**{Pendulum, CartPole, AntBLT, WalkerBLT, HopperBLT}-V.** The "-V" denotes environments that retain velocity-related observations by eliminating position-related observations.

## 9 Implemnetaion Details

We use the PyTorch framework for our experiments. Some basic hyperparameters about the network architectures are listed below

Also, there are some unique parameters in TD3 and SAC, we report them together

Table 1: Environment information

| Environment name | $dim(o)$ | $dim(a)$ | Maximum step |
|---|---|---|---|
| Pendulum-P(position only) | 1 | 1 | 200 |
| Pendulum-V(velocity only) | 1 | 1 | 200 |
| CartPole-P(position only) | 2 | 1 | 1000 |
| CartPole-V(velocity only) | 2 | 1 | 1000 |
| AntBLT-P(position only) | 17 | 8 | 1000 |
| AntBLT-V(velocity only) | 11 | 8 | 1000 |
| WalkerBLT-P(position only) | 13 | 6 | 1000 |
| WalkerBLT-V(velocity only) | 9 | 6 | 1000 |
| HopperBLT-P(position only) | 9 | 3 | 1000 |
| HopperrBLT-V(velocity only) | 6 | 3 | 1000 |
| HalfCheetah-Dir | 17 | 6 | 200 |

Table 2: Hyperparameters

| Hyperparameters | Description | Value |
|---|---|---|
| $\gamma$ | Discount factor | 0.99 |
| lr | Learning rate | 0.0003 |
| $\lambda$ | Balance weight between KL divergence and RL loss | 0.5 |
| $\tau$ | Fraction of updating the target network each gradient step | 0.005 |
| dqn_layers | MLP layer sizes of value function | [256,256] |
| policy_layers | MLP layer sizes of policy network | [256,256] |
| sampled_seq_len | The number of steps in a sampled sequence for each update | 64 |
| batch_size | The number of sequences to sample for each update | 64 |
| buffer_size | The number of saved transitions | 1e6 |
| action_embedding_size | Embedding dimension of action | 16 |
| observ_embedding_size | Embedding dimension of observation | 32 |
| reward_embedding_size | Embedding dimension of reward | 16 |
| gru_hidden_size | Hidden layer size of recurrent encoder | 128 |

## 9.1 Encoding procedure of GRU-ODE

The input observations $x_t$ is the concatenation of observations, actions and rewards, as we mentioned above. The time increment $dt$ is set to a fixed value in regular observation environments. In this paper, we set $dt = 0.1$ for all POMDP tasks.

## 9.2 Integrating with TD3

We use separate recurrent encoders for the actor and critic in order to improve the performance of our method. Owing to the policy network of TD3 updating less frequently than the value function. We modify the iteration equation of the policy network to:

$$\mathcal{L}(\phi_a, \theta, \psi) = - \underset{o \sim \mathcal{D}}{\mathbb{E}} \left[ q_{\psi_1}(o, \pi_\theta(o, z), z) \right] + \lambda_a \mathcal{K}(\phi_a), \tag{12}$$

---

**Algorithm 1** The GRU-ODE algorithm.

---
**Input:** Observations and time difference between observations $(x_t, dt)_{t=1..T}$
$h_0 = \mathbf{0}$
**for** t in 1, 2, …, T **do**
    $\tilde{h}_t = \text{GRUCell}\left(h_{t-1}, x_t\right)$ {Update hidden state}
    $h_t = \text{ODESolve}\left(f_\theta, \tilde{h}_t, dt\right)$ {Solve ODE}
$z_t = \text{MLP}(h_t)$ for all $t = 1..T$
**Return:** $\{z_t\}_{t=1..T}; h_t$

---

Table 3: Hyperparameters of SAC and TD3

| Method | Hyperparameters | Value |
|--------|-----------------|-------|
| SAC | entropy_alpha | 0.2 |
| SAC | automatic_entropy_tuning | True |
| SAC | alpha_lr | 0.0003 |
| TD3 | exploration_noise | 0.1 |
| TD3 | target_noise | 0.2 |
| TD3 | target_noise_clip | 0.5 |

$\mathcal{D}$ denotes the replay buffer. The state $s$ is replaced with observation $o$ owing to the partially observable environments.

The update of the value function is modified to:

$$y(r, o', z') = r + \gamma \min_{i=1,2} Q_{\psi_i, targ}(o', \pi_{\theta, targ}(o'), z'),$$

$$\mathcal{L}(\psi_i) = \mathop{\mathbb{E}}_{(o,a,r,o') \sim \mathcal{D}} \left[ (Q_{\psi_i}(o, a, z) - y(r, o', z'))^2 \right], \tag{13}$$

$$\mathcal{L}(\phi_c, \psi) = \mathcal{L}(\psi_1) + \mathcal{L}(\psi_2) + \lambda_c \mathcal{K}(\phi_c),$$

where subscript $1, 2$ denotes two different critic networks implemented in TD3 and $targ$ denotes the target network. The context variable is computed by $g_\phi(z_t|\tau_{:t})$ as the policy network and value function update, we omit this procedure to simplify the equation.

### 9.3 Integrating with SAC

Similar to TD3, we implement our GRU-ODE in SAC. The training loss of the policy network is modified to

$$\mathcal{L}(\phi_a, \theta, \psi) = - \mathop{\mathbb{E}}_{o \sim \mathcal{D}} \left[ \min_{j=1,2} q_{\psi_j}(o, \pi_\theta(o, z), z) - \alpha \log \pi_\theta(a|o) \right] + \lambda_a \mathcal{K}(\phi_a), \tag{14}$$

and the value function becomes

$$y(r, o', z') = r + \gamma \min_{j=1,2} (Q_{\psi_j, targ}(o', \pi_{\theta, targ}(o'), z') - \alpha \log \pi_\theta(\tilde{a}|o')), \tilde{a} \sim \pi_\theta(\cdot|o'),$$

$$\mathcal{L}(\psi_i) = \mathop{\mathbb{E}}_{(o,a,r,o') \sim \mathcal{D}} \left[ (Q_{\psi_i}(o, a, z) - y(r, o', z'))^2 \right], \tag{15}$$

$$\mathcal{L}(\phi_c, \psi) = \mathcal{L}(\psi_1) + \mathcal{L}(\psi_2) + \lambda_c \mathcal{K}(\phi_c),$$

where symbols have the same meaning as in Eq. 13. The parameter $\lambda$ is set to 0.5 both in TD3 and SAC algorithms.

## 10 ODE Solver Comparison

In this ablation study, we ask two questions in relation to numerical integration. (i) how different numerical solvers influence the final performance of POMDP tasks, and (ii) the effect on the training runtime hours. We experimentally evaluate using different numerical methods as *ODESolve* for integrating $\tilde{h}_t$ in GRU-ODE. From Table 4, we find that there is no obvious relationship between the

Table 4: Final performance of different numerical integration methods

|  | Ant-P | Ant-V | Walker-P | Walker-V |
|------|-------|-------|----------|----------|
| Euler | $1243 \pm 43$ | $998 \pm 93$ | $1487 \pm 81$ | $612 \pm 27$ |
| RK4 | $1183 \pm 67$ | $1045 \pm 112$ | $1405 \pm 64$ | $683 \pm 45$ |
| Heun | $1123 \pm 53$ | $1023 \pm 69$ | $1457 \pm 45$ | $625 \pm 41$ |

complexity of the numerical solvers and the final performance of the PODMP tasks. However, the

RK4 and Heun need much more time for training. We train these methods on a server with NVIDIA TITAN Xp and Intel(R) Xeon(R) CPU E5-2620 v4 @ 2.10GHz as GPU and CPU respectively. The following are rough estimates of average run times for the AntBLT-P environments.

- Ours (with Euler) 12 hours
- Ours (with RK4) 20 hours
- Ours (with Heun) 17 hours

As the results show for context variable coding, the simplest numerical Euler method can work well as *ODESolve* and save training time. We speculate that though the environment is complicated to solve, the context variables always reflect the essential dynamic information of the environments which are more refined than the observations. Thus, simple numerical solvers are enough.

## 11 Time Cost of various baselines

We evaluate the time costs of different baselines on Walker-P environments. From the results, we find our method does not increase the computation a lot compared to other baselines. We analyze that there are two reasons. (I). Within our implementation of ODESolve(), we employ the "Euler" method for computation. This offers substantial reductions in computation time when contrasted with the default "dopri5" approach (Runge-Kutta of order 5 of Dormand-Prince-Shampine) utilized in ODE-RNN. We observe that employing complex ODESolve() methods does not notably enhance performance. Instead, it increases computational time significantly. (II). We are working within the context of RL tasks. When training RL algorithms, it's not just the ODE-GRU that needs training, other networks such as policy networks and value networks also require training. During the training process, a significant amount of time is consumed by sampling transitions from the data buffer and updating the data buffer. Therefore, in terms of the overall training time of our approach, it doesn't significantly exceed the time required by the previous method that used an RNN as an encoder.

Table 5: Time Cost of different methods

| Methods | Time Cost |
|---------|-----------|
| Ours | 12h |
| RMF | 10h |
| VRM | 18h |
| SAC-LSTM | 6h |
| SLAC | 3h |

## 12 Further Performance Comparison

We add two baselines for performance comparison, the RMF (Recurrent Model-free) is a recent SOTA method on POMDP problems and the TD3-FPOW (TD3 with fixed previous observation window) is a modified version of TD3, which takes 3 concatenation frames as the input.

Table 6: Performance comparison

| Environments | SLAC | VRM | SAC-LSTM | RMF | TD3-FPOW | Ours |
|--------------|------|-----|----------|-----|----------|------|
| Ant-P | 950±129 | 1040±75 | 946±47 | 1048±74 | 974±43 | **1243±43** |
| Ant-V | 663±87 | 981±63 | 690±34 | **1021±165** | 903±54 | 998±93 |
| Walker-P | 277±121 | 1121±167 | 971±103 | 1123±176 | 1043±103 | **1487±81** |
| Walker-V | 138±25 | 551±30 | 491±38 | 586±52 | 482±32 | **612±27** |
| Hopper-P | 222±21 | 1851±61 | 1236±39 | 2133±326 | 1654±217 | **2455±87** |
| Hopper-V | 310±41 | **1652±67** | 890±43 | 1495 ± 38 | 1312±87 | 1545±104 |

The result shows that our method achieved the best performance in 4 out of 6 environments.

# 13 Additional ablation study

We study the influence of the shared/separate encoders, and different concatenations of inputs as the ablation study. Due to the time limitation, we conducted these experiments on the Walker-P and Walker-V environments.

Table 7: Performance comparison about shared/separate encoders

| Environments | shared | separate |
|---|---|---|
| Walker-P | 1336 | 1487 |
| Walker-V | 584 | 612 |

The result shows that separate context encoders for the policy network and value function improve the performance.

Table 8: Performance comparison different kinds of inputs

| Environments | o | oa | or | oar |
|---|---|---|---|---|
| Walker-P | 1276 | 1065 | 1396 | 1487 |

The result shows that taking the input $x_t$ as the concatenation of $o_t$, $r_t$ and $a_{t-1}$ for the context encoder (GRU-ODE) improves the performance.

