# OpenReview forum: "ODE-based Recurrent Model-free Reinforcement Learning for POMDPs"
_NeurIPS.cc/2023/Conference — NeurIPS 2023 poster_

### Official Review · Reviewer_kjqZ · 2023-07-06

**Soundness:** 3 good
**Presentation:** 3 good
**Contribution:** 2 fair
**Rating:** 5
**Confidence:** 3

**Summary:**

This manuscript presents GRU-ODE, a new ODE-based recurrent model designed to tackle POMDP control tasks. By utilizing a numerical ODE solver, GRU-ODE effectively incorporates past observations, actions, and rewards into its context. In comparison to previous RNN-based methods, GRU-ODE offers several advantages such as the ability to handle irregular observations and infer unfamiliar physical environments. Experimental results demonstrate the superior performance of GRU-ODE compared to baselines across standard POMDP tasks, meta RL tasks, and tasks with temporally irregular observations.

**After rebuttal**: The authors addressed most of my concerns and the clarification is valid. I have increased my score to 5.

**Strengths:**

1. The experimental results in this work adequately verify the proposed GRU-ODE method, which demonstrates competitive evaluation performance.
2. The paper is well-organized and easy to read.

**Weaknesses:**

1. My main concern is the applicability of the proposed method. The authors assume certain conditions, such as the structure of the reward function and the underlying physics of the task, which may limit the applicability of GRU-ODE. Furthermore, ODE-based encoders fall short in handling discrete game environments such as Atari, as mentioned in line 337.
2. The baselines considered are not sufficiently recent, which weakens the persuasiveness of the evaluation. It is recommended to include results from published papers from the past two years, such as [1, 2].
3. Using ODEs would inevitably incur more training time and inference time. Although the authors compare runtimes with different numerical solvers, it is recommended to also include comparisons to baseline methods.



[1] Ni T, Eysenbach B, Salakhutdinov R. Recurrent model-free rl can be a strong baseline for many pomdps[J]. arXiv preprint arXiv:2110.05038, 2021.

[2] Morad S, Kortvelesy R, Bettini M, et al. POPGym: Benchmarking Partially Observable Reinforcement Learning[J]. arXiv preprint arXiv:2303.01859, 2023.

**Questions:**

1. Could you elaborate on the distinction between the GRU-ODE module and the model used in prior studies [1, 2]?
2. How are baseline methods used for tasks with irregular temporal observations?



[1] Rubanova Y, Chen R T Q, Duvenaud D K. Latent ordinary differential equations for irregularly-sampled time series[J]. Advances in neural information processing systems, 2019, 32.

[2] Lechner M, Hasani R. Learning long-term dependencies in irregularly-sampled time series[J]. arXiv preprint arXiv:2006.04418, 2020.

**Limitations:**

The authors adequately acknowledge the limitations of the proposed method.

---

> ### Author Rebuttal · Authors · 2023-08-08
>
> We thank the reviewer for recognizing the contributions of our paper and giving us constructive suggestions. We will address the reviewer’s concerns in the following parts. We address all of your concerns and show distinctions between our methods with previous works [1]. We hope that our work will receive your recognition and garner positive feedback. We sincerely hope to receive more of your questions and feedback.
>
> **W1**: applicability concern
>
> In our paper, the reward function proposition is employed to provide a theoretical analysis of the rationality behind concatenating both actions and rewards with observations as inputs. Previous works typically validate the rationality of this setting just through experimental results. Actually, the structure of the reward function does not need to be conditioned on such a format. Furthermore, as we mainly focus on continuous control tasks, the underlying physics of such tasks are quite similar, which are generally updated according to the Euler method, thus our method is suitable to apply in such environments. It is true that ODE-based encoders fall short in handling discrete game environments. However, we do not pay much attention to this, our aim is to solve continuous control tasks.
>
> **W2**: performance comparison
>
> As for the baseline problems, we have rerun the method RMF proposed in [2], the comparison results are listed as follows. **Bold** denotes the best performance on each task. Our method holds the best performance in 4 out of 6 environments.
>
> |          | SLAC    | VRM         | SAC-LSTM | RMF          | Ours        |
> | :------- | :------ | :---------- | :------- | :----------- | :---------- |
> | Ant-P    | 950±129 | 1040±75     | 946±47   | 1048±74      | **1243±43** |
> | Ant-V    | 663±87  | 981±63      | 690±34   | **1021±165** | 998±93      |
> | Walker-P | 277±121 | 1121±167    | 971±103  | 1123±176     | **1487±81** |
> | Walker-V | 138±25  | 551±30      | 491±38   | 586±52       | **612±27**  |
> | Hopper-P | 222±21  | 1851±61     | 1236±39  | 2133±326     | **2455±87** |
> | Hopper-V | 310±41  | **1652±67** | 890±43   | 1495 ± 38    | 1545±10     |
>
> It is not fair to compare our method with POPGYM [3], as the environments in POPGYM are almost game and navigation tasks. The control tasks only contain Cartpole and Pendulum, which is similar to the Gym/MuJoCo environment. So we do not compare our method with POPGYM.
>
> **W3**: runtime
>
> For the time cost of concern, we evaluate all the baselines on the Walker-P environments with 1M steps.
>
> | Methods  | Time cost |
> | -------- | --------- |
> | Ours     | 12h       |
> | RMF      | 10h       |
> | VRM      | 18h       |
> | SAC-LSTM | 6h        |
> | SLAC     | 3h        |
>
> **Q1**: distinction
>
> R1: Regarding the distinction, we have made modifications to the computation order of the ODE and GRU updates. This alteration was motivated by the fact that ODE-RNN [1] was originally designed to encode time series backward in time from $t_N$ to $t_0$. Changing the update order has proven beneficial for forward-in-time encoding in our scenario. Additionally, we utilize variational outputs at each step instead of just the last step, and we optimize the KL divergence loss function to regularize the embeddings. Experiments demonstrate that this constraint enhances the agent's performance in the context of RL. In this paper, our primary contribution lies in applying an ODE-based recurrent model to POMDP reinforcement learning environments and putting forward a theoretically sound proposition for combining ODEs with continuous control and partially observable tasks. As shown in Equation (5), we combine continuous action with differential equations, which proposes a method to calculate the action differentiation. As far as I know, this is the first instance of utilizing ODEs within RNNs for POMDPs.
>
> **Q2**: irregular baselines
>
> R2: In irregular tasks, to fairly compare, we assume the incremental $dt$ is known to the agent. In our method, $dt$ is one of the inputs of the $\text{ODESolve}$() function, as the update equation is Equation (2). In the other compared baselines, the incremental $dt$ is added as an additional input, which allows the agent to infer the irregular environments.
>
> [1] Rubanova, Y., Chen, R. T., & Duvenaud, D. K. (2019). Latent ordinary differential equations for irregularly-sampled time series. Advances in neural information processing systems, 32.
>
> [2] Ni, T., Eysenbach, B., & Salakhutdinov, R. (2021). Recurrent model-free rl can be a strong baseline for many pomdps. *arXiv preprint arXiv:2110.05038*.
>
> [3] Morad, S., Kortvelesy, R., Bettini, M., Liwicki, S., & Prorok, A. (2023). POPGym: Benchmarking Partially Observable Reinforcement Learning. *arXiv preprint arXiv:2303.01859*.

---

> > ### Comment · Reviewer_kjqZ · 2023-08-17
> >
> > Thank you for your response. I appreciate the additional information provided. But I still have some questions regarding the proposed method.
> >
> > - I have some doubts about the runtime comparison results. According to [1], the ODE-RNN takes 60% more time than the standard GRU, and the Latent ODE requires roughly twice as much time to evaluate compared to the ODE-RNN. Since there are similarities between the proposed method and ODE-RNN [1], I would appreciate more information on details such as network architectures, model sizes, computation resources, training frameworks, etc., in order to better understand and assess these results.
> > - Could you provide more details on why changing the computation order of the ODE and GRU updates is important? I am not clear on how this change "has proven beneficial for forward-in-time encoding in our scenario". I also notice that forward-in-time encoding has been discussed in [1], which to me seems like a task-specific design choice rather than major model modifications.
> >
> > [1] Rubanova Y, Chen R T Q, Duvenaud D K. Latent ordinary differential equations for irregularly-sampled time series[J]. Advances in neural information processing systems, 2019, 32.

---

> > > ### Author Response · Authors · 2023-08-17
> > > **Replay to Reviewer kjqZ**
> > >
> > > Thank you for your response. Sorry for not solving your confusion in the rebuttal, we hope the following answers can solve them.
> > >
> > > 1. We believe you are an expert in the field of ODEs, and your insights into the computational time issue are highly relevant. First, within our implementation of *ODESolve()*, we employ the "Euler" method for computation. This offers substantial reductions in computation time when contrasted with the default "dopri5" approach (Runge-Kutta of order 5 of Dormand-Prince-Shampine) utilized in ODE-RNN[1]. We observed that employing complex *ODESolve()* methods doesn't notably enhance performance(as shown in Appendix section 10), instead, it increases computational time significantly.  Meanwhile, let us demonstrate that we are working within the context of RL tasks. When training RL algorithms, it's not just the ODE-GRU that needs training, other networks such as policy networks and value networks also require training. During the training process, a significant amount of time is consumed by sampling transitions from the data buffer and updating the data buffer. We believe that models like ODE-RNN do not necessarily require a long time to train for fitting time series data (compared to the training time for RL). Therefore, in terms of the overall training time of our approach, it doesn't significantly exceed the time required by the previous method that used an RNN as an encoder (RMF)[2]. In fact, our network parameters and computational resources have already been listed in the supplementary materials we previously submitted (Section 9 and Section 10). Our network architecture is also identical to the one depicted in Figure 1 and our training procedure follows the setting in [2]. Additionally, we will release our code if the paper is accepted.
> > >
> > > 2. For changing the computation order, we acknowledge that this may seem like a task-specific design. During our experiments, we found that performances of encoders with computational order like ODE-RNN are quite similar to simple GRU encoders (just slightly better) and the visualization results are not reasonable compared to the current version. We analyzed that the distinction between these two orders is which function receives $h_{t-1}$ as the input. We conjecture from the experimental results that using GRU to receive $h_{t-1}$ is much more stable than using *ODESolve()*, as in the POMDP setting inferring unobservable state firstly and then computing continuity can obtain more environment information without losing continuity. The inclusion of the variational process and the utilization of the KL divergence are also seen as modifications to better train the RL algorithm in the POMDP setting. Since our objective is to address POMDP tasks, we reckon that any modifications that can enhance our performance are indeed meaningful.
> > >
> > > [1] Rubanova, Y., Chen, R. T., & Duvenaud, D. K. (2019). Latent ordinary differential equations for irregularly-sampled time series. *Advances in neural information processing systems*, *32*.
> > >
> > > [2] Ni, T., Eysenbach, B., & Salakhutdinov, R. (2021). Recurrent model-free rl can be a strong baseline for many pomdps. *arXiv preprint arXiv:2110.05038*.

---

> > > > ### Comment · Reviewer_kjqZ · 2023-08-21
> > > >
> > > > Thank you for your response. I appreciate the valid points you raised, and I recommend including them in the revised version. As a result, I would like to increase my rating to 5.

---

> > > > > ### Author Response · Authors · 2023-08-21
> > > > > **Replay to Reviewer kjqZ**
> > > > >
> > > > > We will include relevant explanations in the revised version. Thanks for your suggestions. We are pleased to earn your recognition.

---

### Official Review · Reviewer_BSsg · 2023-07-06

**Soundness:** 3 good
**Presentation:** 2 fair
**Contribution:** 2 fair
**Rating:** 6
**Confidence:** 3

**Summary:**

The paper proposes GRU-ODE, which encodes historical observations, actions, and rewards into a latent variable. This model is based on recurrent networks and modifies the GRU topology by separating its memory cell into time-continuous states. The integration of variables is accomplished through neural ODEs. Given the differential process, this formulation can extract more accurate unobservable information from partially observable settings. This approach outperforms baselines in several partially observable and Meta-RL control tasks and irregular partially observable tasks.

**Strengths:**

-	The use of ordinary differential equations (ODEs) allows the model to handle continuous time dynamics, which is a more accurate representation of many real-world systems. This can lead to more accurate and realistic models.
-	The proposed GRU-ODE model is designed to handle partially observable Markov decision processes (POMDPs), a challenging problem in reinforcement learning. By encoding historical observations, actions, and rewards into a latent variable, the model can extract unobservable information from the environment. I found this quite interesting specifically, the results presented in Fig~3 are quite convincing.
-	The paper demonstrates that the proposed method is robust against irregular observations, which is a significant advantage in real-world applications where observations may not always be regularly or uniformly sampled.
-	The authors show that their approach outperforms baselines in several regular partially observable control tasks.
-	Its nice to see that the encoder and the policy network together from the same data buffer. Simplifies the training process.

**Weaknesses:**

-	Because networks were trained from the same data buffer – does this cause exploration problems? Potentially why this approach is unable to handle sparse rewards?
-	The models performance should be tested on robotics tasks e.g., Franka kitchen(https://github.com/Farama-Foundation/D4RL/tree/master/d4rl/kitchen/third_party/franka). It's unclear how well the model would generalize to other slightly more complicated dynamics.
-	The model makes several assumptions, such as the use of ordinary differential equations (ODEs) to model the dynamics of the system. If these assumptions do not hold in a particular application, the model's performance will be impacted i.e., the stationarity and time-dependence.

**Questions:**

-	Is this the first combination of GRU-ODE – or the first applied to control?
-	Could you compare this recent SOTA POMDP formulations e.g., Dreamer with the same partial inputs?
-	How different is the performance if the same latent context variable (i.e., same GRU-ODE encoder) is used for both the action and critic?
-	Currently, only positions / velocities are removed? How does performance change if multiple observation information is removed. Would it be possible to evaluate this for some environments?
-	For $x_t$ why does it have to be a concatenation of $o_{0:t}, a_{0:t}$ and $r_{0:t}$? How much does the performance vary when you remove this extra signal?
-	How is the incremental $dt$ varied in the numerical ODE solver for irregular observations? If the $\delta = 0.05$ is varied, does it cause issues when the time is different between the generative process and model?
-	Can you relax the deterministic policy function assumption? Would this cause issues with differentiating $a(t)$ wrt to time.
-	How is discount formulated?
-	How does this formulation perform in stochastic settings?
-	Would it be include a metric that ranks performance conditioned on the time budget?
-	Was there any pretraining?

**Limitations:**

-	The authors have adequately addressed the concerns about the  Model Complexity. This is because the proposed GRU-ODE model, while powerful, is likely more complex than traditional models. This could make it more computationally intensive and harder to implement.
-	Furthermore, these setting are relevant for lower-level control e.g., the Hopper tasks. However, for environments with discrete dynamics these might not be relevant.

---

> ### Author Rebuttal · Authors · 2023-08-08
>
> We thank the reviewer for recognizing the contributions of our paper and giving us constructive suggestions. We have endeavored to address all of your inquiries and provide explanations for the issues raised in the weaknesses section. We aspire to resolve your concerns about assumptions and ablation studies and earn your recognition for the contributions of our paper. We look forward to receiving your feedback and further questions.
>
> **W1**: exploration and sparse reward
>
> In our experiment, there is no exploration problem, as we use the Gaussian network for policy learning. From our perspective, rewards are crucial information that contains relevant details about states and actions in POMDP settings. Additionally, since we use recurrent neural networks as encoders, sparse rewards may dissipate dependence over long iterations.
>
> **W2**: robotic tasks
>
> For further experiments, we train our network in the Kitchen-Burner environment[1]. After training for 100,000 steps, our reward reached around 1.2, which is competitive with relevant methods at the same steps. However, we found that methods such as SAC perform badly. Since we do not utilize demonstrations to guide agent learning, direct training methods are expected to perform less effectively compared to demonstration-guided RL approaches. We believe that, combined with demonstration, our approach can be generalized to relevant tasks.
>
> **W3**: assumption
>
> For continuous control tasks, almost all the physical engines update the state according to the Euler method, which follows the ordinary differential equations. As our aim is to solve continuous control tasks, we reckon that our method can be used in most environments.
>
> **Q1**: first proposal
>
> R1: We are the first to apply the ODE-based recurrent model to RL and robotic tasks, particularly in the context of POMDP settings.
>
> **Q2**: comparison
>
> R2: With our understanding, the dreamer is applied to pixel input and is not used for POMDP, which is not suitable for our baseline. We have compared a recent SOTA method RMF [2]. **Bold** denotes the best performance on each task.
>
> |  | VRM| RMF|Ours|
> | -| -| -| -|
> | Ant-P| 1040±75 |1048±74|**1243±43**|
> | Ant-V| 981±63| **1021±165**|998±93|
> | Walker-P |1121±167| 1123±176| **1487±81** |
> | Walker-V |551±30 | 586±52|**612±27**|
> | Hopper-P |1851±61| 2133±326| **2455±87** |
> | Hopper-V | **1652±67**| 1495 ± 38|1545±104|
>
> **Q3**: shared/separate encoders
>
> R3: We have compared that using the shared and separate encoders, as shown in the table below. Shared and separate recurrent actor-critic architectures have been compared and show a huge performance gap [2]. In our method, we do not find such big gaps, but there are still some performance gaps.
>
> | |shared|seprarate|
> |-|-|-|
> |Walker-P|1336|1487|
> |Walker-V|584|612|
>
> **Q4**: positions/veloicties
>
> R4: Actually, in the POMDP setting, the state space is divided into two parts, the positions, and the velocities. You can also regard the experimental environment as having only position/velocity information, which is equivalent to the removal of velocity/position information. Thus, multiple observation information has been removed in this paradigm.
>
> **Q5**: input influence
>
> R5: There is a misunderstanding, the input of our encoder is the concatenation of $o_t$,$a_{t-1}$, and $r_t$. We do not take the whole historical information as input. As our proposition, the action $a_t$ and reward $r_t$ could help to infer the unobservable observations, previous works also mentioned that containing action and reward as input could improve the performance[2]. We do ablation study in four scenarios, $[o_t]$,  $[o_t,a_{t-1}]$, $[o_t,r_t]$, and $[o_t,a_{t-1},r_t]$ in Walker-P environments.
>
> |  |o|oa|or|oar|
> |-|-|-|-|-|
> | Walker-P|1276±134|1065±65|1396±74|1487±81|
>
> **Q6**: $dt$ variation
>
> R6: In irregular setting, $dt$ follows $dt\sim\mathcal{U}\left(0,2\delta\right]$. Only encoders take $dt$ as input, so there is no inconsistency in our model. $\delta=0.05$ is the default setting, it could be changed to any rational value.
>
> **Q7**: deterministic assumption
>
> R7: The deterministic policy assumption is for the sake of rigor. This assumption is still valid in a stochastic setting. For the stochastic policy,  the probability distribution of actions is computed from the state first, and then actions are sampled from this distribution. In this process, the former part is more critical. This part is differentiable and aligns with our assumption. However, the latter half is non-differentiable and does not align with our assumption. Therefore, if $a(t)$ is a random value represented by the probability distributions, our assumption is still valid. However, if $a(t)$ represents the sampled actions, this assumption needs to be revised. Overall, because Gaussian random policy networks are commonly used in modern networks to learn action probability distributions, assuming $a(t)$ is the output of the network, this assumption still holds for stochastic policies.
>
> **Q8**: discount factor
>
> R8: The discount factors keep 0.99 across all the experiments.
>
> **Q9**: stochastic setting
>
> R9: The formulation does not need to be changed in the stochastic setting as explained in R7.
>
> **Q10**: time rank
>
> R10: It is hard to design an evaluation index that both concerns time cost and performance, maybe a figure is meaningful, but we could not show it here. We list the time cost according to the performance in the Walker-P environment.
>
> |   | Time Cost | Performance |
> |-| -| -|
> |Ours|12h|1487±81|
> |VRM|18h|1121±167|
> |RMF|10h|982±339|
> |SAC-LSTM|6h| 971±103|
> |SLAC|3h|277±121|
>
> **Q11**: pretrain
>
> R11: There is no pretraining.
>
> [1] Hakhamaneshi, K., Zhao, R., Zhan, A., Abbeel, P., & Laskin, M. (2021). Hierarchical few-shot imitation with skill transition models. *arXiv preprint arXiv:2107.08981*.
>
> [2] Ni, T., Eysenbach, B., & Salakhutdinov, R. (2021). Recurrent model-free rl can be a strong baseline for many pomdps. *arXiv preprint arXiv:2110.05038*.

---

> > ### Comment · Reviewer_BSsg · 2023-08-21
> >
> > Thank you for your response and including the additional experiments. I believe the approach has merit (despite the concerns about computational constraints) and as such I will be increasing my score by 2 points.

---

> > > ### Author Response · Authors · 2023-08-21
> > > **Replay to Reviewer BSsg**
> > >
> > > I am pleased that we resolved your issues and received your acknowledgment of our paper.

---

### Official Review · Reviewer_6E9a · 2023-07-06

**Soundness:** 3 good
**Presentation:** 1 poor
**Contribution:** 2 fair
**Rating:** 6
**Confidence:** 3

**Summary:**

The authors propose a neural ODE-based GRU for use in partially observable RL. They effectively "filter" GRU recurrent states using an ODE gate, before feeding the filtered state back into the GRU. They use a variational approach, where their model $g$ spits out a mean and variance. They find that training with just reward results in a highly-variable latent state $z_t$, so they constrain $z_t$ via an additional KL loss term with the prior recurrent state.

They go on to evaluate their method across partially observable and meta RL tasks. They examine cartpole with masked velocity/position, as well as various PyBullet/MuJoCo masked control tasks, comparing against SAC+LSTM and a few other methods. They go on to evaluate tasks with irregularly-spaced observations in time. The idea is that a continuous temporal representation using the ODE is more amenable to continuous time than a discrete RNN.

**Strengths:**

- Their method is novel -- I was not able to find other papers using ODEs within RNNs for POMDPs.
- Their half-cheetah experiment is very promising, showing that their method matches oracle performance while others cannot
- The paper is generally well written

**Weaknesses:**

- Although their method seems interesting, it seems impractical given current hardware. They must solve an ODE at each timestep (even during inference). Given that interesting RL tasks require collecting tens/hundreds/thousands of millions of transitions, their method is likely to be restricted to toy tasks, at least for now.
- Many of their experiments are limited to 1M frames, which seems low. This is likely due to the high computational complexity of their method. For most environments, it is not clear that their conclusions would hold at 10M or 50M steps as their policies do not appear to have converged. In their defence, the half-cheeta experiment appears very promising, and the Ni et al. paper they reference trains the PyBullet tasks for the same number of environment steps.
- They do not ablate components of their approach

**Questions:**

- If you are going to modify the internals of a GRU, I think it makes sense to write out the full GRU equations.
- "We found that the reward signals could help the agent learn unobservable information, leading to improve performance of the agent in PO environments."
    - What does this mean? What is "unobservable" information in this context? Generally POMDP observations are modeled as $o \sim O(s)$, in that the state is always indirectly observed.
- Line 135/136: Are you providing the true Markov state to your model via the loss function? This limits your approach to tasks where we know the underlying state, like toy tasks. After reading the training section, I don't think this is the case but it might confuse readers.
- I would be interested in further information/experiments about highly-variable recurrent states and how a KL regularizer helps
    - I do also wonder how responsible this is for your results. How much is the ODE that is helping compared to the variational state constraints.

**Limitations:**

- The authors claim that their method performs well in discrete tasks, such as atari. This is to be expected, as a continuous-time representation does not make sense in these scenarios.
- They also find that solving ODEs is computationally expensive

---

> ### Author Rebuttal · Authors · 2023-08-08
>
> We thank the reviewer for recognizing the positive aspects of our paper, and we will address the reviewer’s concerns in the following parts.
>
> **W1**: computation
>
> We agree with your statement that calculating ODE at each step requires additional computational resources. However, we would like to clarify that in our paper, we primarily used the Euler method to approximate the solution of the differential equation. Compared to not performing ODE iterations, this method does not significantly increase the computational complexity. Therefore, we believe that there is potential for this work to be extended to more environments.
>
> **W2**: frame limitation
>
> About the number of frames, not only us, all the baselines we compare are limited to about 1M frames. According to the time cost, running 10M or 50M frames takes a long time for all methods.
>
> **W3**: ablation components
>
> We add some ablation studies to answer Q4. Due to time constraints, we only conducted experiments in some environments and listed typical problems.
>
> **Q1**: GRU equations
>
> R1: There are some errors in Openreview that do not display the equations correctly, which we will directly add the equations in the appendix.
>
> **Q2**: "unobservable" meaning
>
> R2: In this context, the term 'unobservable' refers to the velocity/position information that the agent cannot directly obtain. This information is not contained in the observation $o$. Previous research has demonstrated that adding reward signals to inputs can improve performance. As discussed in our paper, the reward function typically consists of $r(s, a)=d(s,s^*)+c||a||_2$, which can be considered as computed from the state $s$ and action $a$ (where the observation $o$ represents a partial value of the state $s$). Therefore, in reverse, if the reward $r$, action $a$, and observations $o$ are known, the unobservable state can be inferred through computation.
>
> **Q3**: observation in the loss function
>
> R3: Sorry for our writing makes you misunderstanding, we provide the observation instead of the state. In lines 135/136, we aim to show that the reward signals consist of action and state, as the observations represent a partial value of the state, the agent could infer the unobservable information from reward, actions, and observations. We will modify  the sentence into "As the reward function typically consists of $r(s, a)=d(s,s^*)+c||a||_2$, where state $s$ is implicitly measured by the environments and can not be fully observed."
>
> **Q4**: ablation study
>
> R4: We remove the variational process for the ablation study. During the training process, there are significant performance oscillations. The best performance is good, but the average is not very high (in Walker-P the performance is 1246±193). Then, we only ablated the KL divergence constraint in the loss function and found that the experimental results depended on the environment. In the Walker-P environment, the drop in the average performance was not significant, but the variance increased substantially (1355±201). In the Hopper-P environment, the mean performance was greatly affected (1931±176). Due to time constraints, we were unable to conduct more experiments on other environments. However, based on the current situation, we believe that adding the KL divergence constraint and the variational process can better constrain the learning of the context variable. For comparing ODE with the variational state. We remove the ODE process and variational process to do an ablation study, the performances are 1057±78 and 1246±193 respectively in the Walker-P task, which shows the significant roles of both processes in our method.

---

> > ### Comment · Reviewer_6E9a · 2023-08-17
> >
> > What if you were to apply the KL divergence regularization to the other methods like SAC-LSTM/SLAC/VRM? Would they outperform your ODE-based approach?

---

> > > ### Author Response · Authors · 2023-08-17
> > > **Replay to Reviewer 6E9a**
> > >
> > > Thank you for your reply.
> > >
> > > Let me first give you our conclusion, these three methods will not outperform our method. Actually, the SLAC and VRM methods use the evidence lower bound (ELBO) as part of the optimization target, the ELBO can be written as:
> > >
> > > $\textit{ELBO}(q)=\mathbb{E}_q[\log p(x|z)]-\textit{KL}(q(z)||p(z))$,
> > >
> > > which contains the KL divergence. For the SAC-LSTM, at the beginning of our research, we tried to combine this algorithm with KL divergence optimization. However, we found the performance is just slightly improved and even inferior to VRM. Thus, all these three methods will not outperform our method with KL divergence.

---

> > > > ### Comment · Reviewer_6E9a · 2023-08-18
> > > >
> > > > So that ELBO loss pushes all recurrent states to be unit-normal. In your Eq. 8, it is my understanding that you are enforcing a different constraint: the current state must be similar the previous state, right?
> > > >
> > > > > We set the prior to our previous posterior
> > > >
> > > > This seems like a stronger and more useful constraint, as it ensures that the state's rate of change remains low, where as the ELBO loss says nothing about how quickly the state can change. With your SAC-LSTM experiments, was the prior based on the previous state?

---

> > > > > ### Author Response · Authors · 2023-08-19
> > > > > **Replay to Reviewer 6E9a**
> > > > >
> > > > > Thank you for your further reply. We have revised our response because some of our understandings were not clearly conveyed before.
> > > > >
> > > > > Yes, we use KL loss with the prior based on the previous state in SAC-LSTM and our aim is to avoid significant changes between hidden states of adjacent time steps. Actually, at first, we found the hidden state of the LSTM does not change hugely (we regard the computation of hidden state update as simple). However, with our GRU-ODE, due to the unavailability of a supervised learning framework, there were instances where the hidden state exhibited significant changes, especially when compared to LSTM in the context of RL. Within the RL framework, where the GRU-ODE serves as an encoder to represent historical information, rapid changes between the context variables $z_t$ may compromise its representational capacity and potentially affect the performance of the RL algorithm. Thus, we made this improvement and used KL divergence to regularize the adjacent outputs. You could find some similar regularization methods in [1,2].
> > > > >
> > > > > [1] Zintgraf, L., Shiarlis, K., Igl, M., Schulze, S., Gal, Y., Hofmann, K., & Whiteson, S. (2019). Varibad: A very good method for bayes-adaptive deep rl via meta-learning. *arXiv preprint arXiv:1910.08348*.
> > > > >
> > > > > [2] Deng, F., Jang, I., & Ahn, S. (2022, June). Dreamerpro: Reconstruction-free model-based reinforcement learning with prototypical representations. In International Conference on Machine Learning (pp. 4956-4975). PMLR.

---

> > > > > > ### Comment · Reviewer_6E9a · 2023-08-20
> > > > > >
> > > > > > Thanks for the clarification. To be fully convinced, I would need to see ablations for the previous-state-prior KL constraint applied to the other methods. It seems unfair to apply the previous-state-prior KL constraint to your method and not the others, as it is a general method and not specific to the ODE-based approach. As such, I will keep my score a weak accept.

---

> > > > > > > ### Author Response · Authors · 2023-08-21
> > > > > > > **Replay to Reviewer 6E9a**
> > > > > > >
> > > > > > > Thank you for your response and overall recognition of our paper. Your thorough approach to ablation studies has been highly beneficial to us. In our future research, we intend to conduct more rigorous ablation studies.

---

### Official Review · Reviewer_FbJ4 · 2023-07-06

**Soundness:** 3 good
**Presentation:** 3 good
**Contribution:** 3 good
**Rating:** 6
**Confidence:** 4

**Summary:**

The author present a method for doing deep reinforcement learning for POMDPs with uneven time between observations. Their method uses TD3 with Neural-ODE+GRU encoders for the actor and the critic. They test on various partially observed control tasks, as well as tasks with uneven observation times.

**Strengths:**

The approach is sound and the results are good even though limited. To the best of my knowledge this is a novel approach for dealing with continuous time observation in RL and generally for using Neural-ODEs in RL. The problem of dealing with non-uniform time between observation is interesting for a lot of problems, especially robotics. The presentation is good (except a few minor things which I will highlight below). I find the integration of continuous-time actions with equation (5) clever, I would have gone with a constant action in between timesteps if I was going to tackle this problem. There will be a lot of small complaints below, but keep in mind that my overall opinion of the paper is positive.

**Weaknesses:**

I think the evaluation environments are a bad choice for POMDPs. Just removing part of the state is a very basic way to simulate partial observability. In the case of observing position and not velocity, think of all of the typical control benchmarks (Deepmind control suite for example) with pixel observations. They're equivalent (you can't observe velocities from a single frame) and yet people mostly treat them as MDPs and ignore the partial observability. How? Because you simply have to stack a couple of frames and use regular deep RL approaches. The partial observability aspect is easily addressed and is simply not interesting. There's no information gathering happening. This is a wider complaint because that's the way that a lot of POMDP papers do evaluation, so I won't take off to many points for this. But as a research community, we really need to sit down and find better benchmarks for POMDPs which highlight their ability to combine information gathering and reward maximization, which is the beauty of the framework.

The way I read the first result section is this way: we do well on the easy problem, position only (but I'd like to see basic TD3 with a concatenation of 3-4 previous observations as a baseline), but on the harder, actually interesting problem, we do just as bad as the other approaches.

I find the use of non-standard deprecated environments odd, even though you mention they're "more difficult" (how so?).

Because all of the evaluation seems to be on non-standard environments, it's difficult to assess the significance of the results. My intuition is that pendulum from position only is actually easily solved, so it raises suspicion that's you are able to beat everyone else by a large margin. No information on how other algorithms were tuned was provided. Convergence speed, which is what your algorithm seems to be better at, varies wildly depending on hyperparameters.

Some sentences I found hard to read: line 201 to 203, 170. 52-54.

Line 56 is a bit hand-wavy. What do you mean by extract more accurate unobservable information? What the experiments show is faster reward maximization. No evaluation of the method's ability to gather information is shown.

The sentence about POMDP starting at line 107 is not correct. You're confusing meta-RL and POMDPs. Please give a formal mathematical definition of a POMDP and provide a citation. You will see that for most the the literature the reward function is the same across episodes. Yes, you can transform a meta-RL problem into a POMDP, by adding a hidden task variable as input to the reward function.

Equation (2) $x_t$ is defined too far from it's appearance in the equation, I was confused at first

(minor) Line 178, the comment about "physical principles" feels a bit hand-wavy to me. The type of Neural ODEs that are used here are no more physically inspired than typical RNNs; no physics knowledge is given to them. Yes a lot of mechanics is usually written down by engineers and physicists as ODEs, but I don't think that means Neural ODEs have "underlying physical principles", unless some actual physics guide the learning.

**Questions:**

Why are there no experiments on pixels observations? It seems the algorithm is scalable.

Why did you use the deprecated environments? What makes them harder and why does that matter as long as the comparison is fair?

Model implementation, item 2 and 3. Why do you add back o_t as input to the network. It's already encoded by g, no?

**Limitations:**

In their limitation section, the authors mention atari. I don't think this is a very important limitation; it's OK, nobody expected ODEs to do well here. What worries me more is the ability to scale to pixels at all. They mention that their method is slower and I suspected that.

---

> ### Author Rebuttal · Authors · 2023-08-08
>
> We thank the reviewer for recognizing the positive aspects of our paper, and we will address the reviewer’s concerns in the following parts.
>
> **W1**: POMDP setting
>
> We agree with your opinion about the POMDP framework, during our experiments, we found almost all the baselines are proposed on position/velocity only environments, so it is harder for us to compare in other benchmarks.
>
> **W2**: TD3 baseline
>
> We add the basic TD3 method with a fixed partial observations window (TD3-FPOW, the window size is 3) as the input and compare it with our methods. We also compare a recent SOTA method RMF [1] here. Due to time constraints, we conducted experiments only on a subset of environments.
>
> |   |  Ours   |   VRM    | TD3-FPOW |   RMF    |
> | :------: | :-----: | :------: | :------: | :------: |
> |  Ant-P   | **1243±43** | 1040±75  |  974±43| 1048±74  |
> |  Ant-V   | 998±93  |  981±63  |  903±54  | **1021±165** |
> | Walker-P | **1487±81** | 1121±167 | 1043±103 | 1123±176 |
>
> **W3**: deprecated environment
>
> The reason we used PyBullet as our experimental environment is that the baseline we are comparing against, such as VRM, was evaluated on this environment (specifically, it was evaluated on the roboschool environment, which has now been updated to PyBullet). We believe that this setting makes the comparison more fair because it eliminates the issue of adapting the baseline to different environments.
>
> **W4**: evaluation problem
>
> In fact, the baselines we compared against were all open-source codes from relevant papers. In the pendulum experiment, our method achieves similar performance to VRM after the models converge, but with a faster convergence speed. We believe this is reasonable because the structure of the VRM encoder is similar to an autoencoder, which requires encoding and decoding of inputs, thus needing more steps to optimize the encoder. Although we have ODE steps, it is still faster than the decoding procedure.
>
> **W5**: sentence modification
>
> We will modify lines 201 to 203 into “Compared to modeling the transition or reward function directly using the network, learning distribution embeddings need fewer parameters which makes inference easier.”
>
> Line 170 into "Assuming the increment $dt$ in the ODEs equals the observation interval,"
>
> Line 52-54 "At each time step, the model first iterates using the conventional GRU equations and then computing the differentiation based on the ODEs."
>
> The description on line 56 may not have been precise enough in explaining the unobservable information. Our intention was based on our visualization results and experimental performance, which led us to believe that our model can better extract unobservable information. As in the visualization results, similar values were mostly clustered together.
>
> **W6**: POMDP definition
>
> The definition of pomdp is "Typical POMDPs are used to describe decision or control problems in which the underlying states of the environments cannot be directly observed. A POMDP is a tuple $(\mathcal{S}, \mathcal{A}, \mathcal{O}, \mathit{T}, \mathit{O}, \mathit{R}, \mathit{\gamma})$. $\mathcal{S}$ is a set of states, $\mathcal{A}$ is a set of actions, and $\mathit{T}: \mathcal{S}\times\mathcal{A}$ is the state-transition probability function. Let $\mathcal{O}$ be the observation set and let $\mathit{O}: \mathcal{S}\times\mathcal{A}\times\mathcal{O}\to[0,1]$ be the observation probability. The reward function $\mathit{R}:\mathcal{S}\times\mathcal{A}$ decides the reward during state transition and $\mathit{\gamma}$ is the discount factor. The objective is to learn a policy function to maximize expected discounted rewards $\mathbb{E}\left[\sum\limits_{t=0}^{T-1}\gamma^tr_{t+1}\right]$ with a finite horizon $T$.
>
> **W7**: notation definition
>
> We will add "$x_t$ is the concatation of $o_t$, $a_{t-1}$ and $r_t$ " after the Equation (2)
>
> **W8**: physical principle
>
> We have considered your views carefully. We acknowledge that the "underlying physical principle" is not clear in ODE, but we still reckon Neural ODE could help to learn the underlying dynamic as the intrinsic structure of Neural ODE contains the differentiation process.
>
> **Q1**: pixel observations
>
> R1: Because our primary focus is on POMDP problems, there is no standard paradigm and baseline for POMDP problems in image/pixel observation environments. We attempted to use a two-layer convolutional network as the image encoder, but the results were unsatisfactory. We believe that the representation ability of the image encoder significantly affects the performance. Combining some image encoding methods may help our method scale to the pixel observation domain.
>
> **Q2**: environment setting
>
> R2: I apologize for any misunderstanding caused by our writing. The environment we used, Pybullet, is not deprecated. We chose Pybullet as our experiment baseline because VRM, which we are comparing against, was initially evaluated on roboschool. However, roboschool has been deprecated and updated to Pybullet. To compare consistently, we selected Pybullet as our experimental environment. The "harder" means PyBullet is harder than MuJoCo, as its observation space contains robots’ coordinates, joint angles, and velocities.
>
> **Q3**: model implementation
>
> R3: Yes, the observation $o_t$ is already encoded by GRU-ODE $g()$. However, the GRU-ODE $g()$ is regarded as a context encoder in our experiment, which output a context variable $z_t$. According to the definition of context-based RL, the context variables are regarded as auxiliary inputs to help the policy and value networks better calculate action distributions and state-action values. Thus, $g()$ is an auxiliary encoder, which does not modify the framework of the policy network and value function. Also, in this context-based paradigm, the encoder and RL algorithm are more flexible to revise.
>
> [1] Ni, T., Eysenbach, B., & Salakhutdinov, R. (2021). Recurrent model-free rl can be a strong baseline for many pomdps. *arXiv preprint arXiv:2110.05038*.

---

> > ### Comment · Reviewer_FbJ4 · 2023-08-16
> > **Some notes on the authors' reply**
> >
> > W6: OK, is this going to be changed in the paper?
> > Q1: Dreamer, DreamerV1 are all well known RL methods that have recurrent memory in them and can deal with POMDPs. They perform well in pixel-based environments.

---

> > > ### Author Response · Authors · 2023-08-17
> > > **Replay to Reviewer FbJ4**
> > >
> > > Thanks for your reply.
> > >
> > > About **W6**: Yes, we will add this definition to our paper and delete the previous version.
> > >
> > > About **Q1**: Upon revisiting the relevant literature, we acknowledge our oversight and apologize for the misunderstanding in our initial response. However, upon closer examination, methods like Planet[1] or Dreamer[2] define POMDP as "Since individual image observations generally do not reveal the full state of the environment, we consider a partially observable Markov decision process (POMDP)," where images themselves are treated as partially observable states. This distinction sets it apart from our experimental setup. Consequently, we consider these two approaches to exist on different baselines. We will clarify the distinction between these environment settings and provide references to the relevant pixel-based methods in the related work section. The challenge of learning tasks in pixel-based environments lies in improving image representation ability, an aspect we intend to address in our future works. We are thankful for the valuable insights you've provided.
> > >
> > > [1] Hafner, D., Lillicrap, T., Fischer, I., Villegas, R., Ha, D., Lee, H., & Davidson, J. (2019, May). Learning latent dynamics for planning from pixels. In *International conference on machine learning* (pp. 2555-2565). PMLR.
> > >
> > > [2] Hafner, D., Lillicrap, T., Ba, J., & Norouzi, M. (2019). Dream to control: Learning behaviors by latent imagination. *arXiv preprint arXiv:1912.01603*.

---

### Author Rebuttal · Authors · 2023-08-10

We are incredibly grateful for the positive assessment and highly detailed and constructive feedback on our paper. In this rebuttal, we have addressed the comments of the review team. This has led to multiple improvements including a better exposition of the proposition, more complete ablation studies, and new applications. Below we address each question and minor questions point by point. Due to character limitations, we have simplified the tables in the rebuttal. The complete tables can be referenced in the PDF file. We hope that our contributions will receive your positive evaluation.

---

### Decision · Program_Chairs · 2023-09-21

**Decision:**

Accept (poster)

**Comment:**

This paper tackles the problem of partial observability in deep reinforcement learning (RL). The authors introduce a method combining recurrent neural networks (GRU) and Neural ODE for the encoder of actor/critic networks.
The paper initially received mixed reviews, with two accept and two reject recommendations. The main concerns pointed out by reviewers related to the novelty of the approach, its applicability for more complex control/robotics tasks and for partial observability including perception (e.g. from pixels), its scalability due to the the Neural-ODE's cost, and the experimental comparison with stronger and more recent baselines. The rebuttal resolved the most important points raised by the reviewers, who were inclined to accept the paper.

The AC has thoroughly reviewed the submission and the discussions. The problem of partial observability is central in robotics and RL. Although the approach may seem incremental given the extensive attempts for using neural-ODE in dynamical models, the combination N-ODE-GRU for improving POMDP solutions open a promising paths. It could inspire researchers for improving the current limitation of the works, e.g. its scalability with advanced solvers, or including perception for more advanced robotic tasks. The rebuttal also clarifies several points regarding experiments.
Therefore, the AC recommends acceptance but highly recommends the authors to carefully take into account the remarks of the reviewers when preparing the final paper.